

# Terrestrial wildlife as indicators of microplastic pollution in western Thailand

Jiraporn Teampanpong[1],* and Prateep Duengkae[2],*

[1] Department of Conservation, Faculty of Forestry, Kasetsart University, Bangkok, Thailand
[2] Department of Forest Biology, Faculty of Forestry, Kasetsart University, Bangkok, Thailand
* These authors contributed equally to this work.

## ABSTRACT

Plastic pollution in terrestrial wildlife represents a new conservation challenge, with research in this area, especially within protected areas (PAs), being scant. This study documents the accumulation of microplastics (MPs) in terrestrial wildlife both inside and outside PAs in western Thailand. Carcasses of road-killed vertebrates in good condition, as well as live tadpoles, were collected to examine their exposure to plastic pollution. The digestive tracts of the vertebrate carcasses and the entire bodies of tadpoles were analyzed for MPs, which were identified if they measured over 50 μm. A total of 136 individuals from 48 vertebrate species were examined. The sample comprised snakes (44.12%), birds (11.03%), lizards (5.15%), tadpoles (32.25%), amphibians (5.88%), and mammals (1.47%). In total, 387 MPs were found in 44 species (91.67%), with an average occurrence of 3.25 ± 3.63 MPs per individual or 0.05 ± 0.08 MPs per gram of body weight. The quantities of MPs significantly varied among the animal groups, both in terms of number per individual ($p < 0.05$) and number per gram of body weight ($p < 0.01$). Furthermore, a significant difference in MP quantities was observed between specimens collected inside and outside PAs on an individual basis ($p < 0.05$), but not on a body weight basis ($p = 0.07$). Most MPs were fibers (77%), followed by fragments (22.22%), with only a minimal presence of film (0.52%) and foam (0.26%). Of all the MPs identified, 36.84% were confirmed as plastics or fibers made from natural materials, and 31.58% were plastics, including Polyethylene (PE), polyethylene terephthalate (PET), polypropylene (PP), Polyvinylidene chloride (PVDC), and polyester (PES). Additionally, fibers made of cotton, and those containing polyurethane (PU), rayon, PES, and combinations of rayon and PU, were identified. The quantities of MPs were significantly influenced by animal body weight, factors associated with human settlement/activity, and land use types. Our findings highlight the prevalence of plastic pollution in terrestrial vertebrates within Thai PAs. Further toxicological studies are required to establish plastic pollution standards. It is proposed that snakes, obtained from road kills, could serve as a non-invasive method for monitoring plastic pollution, thus acting as an indicator of the pollution threat to species within terrestrial ecosystems. There is an urgent need for the standardization of solid waste management at garbage dump sites in remote areas, especially within PAs. Conservation education focusing on MP occurrence, potential sources, and impacts could enhance awareness, thereby influencing changes in behaviors and attitudes toward plastic waste management at the household level.

Corresponding author
Jiraporn Teampanpong,
jiraporn.tea@ku.th

## INTRODUCTION

The production and use of plastic have surged due to its durability, affordability, light weight, flexibility, and resistance to corrosion (*Zhang et al., 2020a*). Despite its potential for biodegradability, the decomposition of plastic waste can be slow, leading to its mismanagement and consequential environmental pollution. This pollution impacts society, the economy, and the quality of life (*Baho, Bundschuh & Futter, 2021*). Environmental factors such as sunlight, wind, waves, and microorganisms gradually fragment large plastic items into debris and microplastics (MPs), defined as particles sized between 100 nm and 5 mm (*Ng et al., 2018*). Due to their small size, MPs pose a significant challenge for removal from the environment and can disperse across atmospheric, aquatic, and terrestrial settings (*Prokić et al., 2021*), becoming bioavailable to a broad range of organisms (*Wang, Ge & Yu, 2020*). Organisms accumulate MPs through ingestion, inhalation, or trophic transfer. Numerous studies have documented the effects of MPs on various organisms within different ecosystems and their functions (*Baho, Bundschuh & Futter, 2021*). The accumulation of MPs can lead to false satiation, resulting in malnutrition, an insufficient energy supply (*Windsor et al., 2019*), impacts on somatic growth and/or metamorphosis (*Messinetti et al., 2018*), alterations in metabolic rates (*Choi, Hong & Park, 2020*), and ultimately mortality (*Windsor et al., 2019*). In vertebrates, MP accumulation can disrupt intestinal functions, inducing oxidative stress and inflammation, altering intestinal permeability and mucus expression and volume, shifting gut microbiota composition, and destabilizing the intestinal environment, which can lead to the recruitment of immune cells (*Huang et al., 2021*). Therefore, terrestrial vertebrates exposed to plastic pollution levels may experience alterations in fundamental physiological and ecosystem processes (*de Souza Machado et al., 2018*), making MPs an emerging threat to biodiversity in natural and semi-natural ecosystems (*Zang et al., 2020*) and a global emerging pollutant (*Richardson et al., 2023*), causing adverse ecological surprises (*Baho, Bundschuh & Futter, 2021*).

In recent decades, the scientific community has increasingly focused on studying MP occurrences, predominantly in marine environments (*Barboza & Gimenez, 2015*) over freshwater (*Li, Liu & Chen, 2018*; *Windsor et al., 2019*) and terrestrial habitats (*de Souza Machado et al., 2018*; *Rillig & Lehmann, 2020*). MP research has broadened to encompass biotas, with a greater emphasis on marine (*Ugwu, Herrera & Gómez, 2021*) than on freshwater (*Cera & Scalici, 2021*) and terrestrial (*Rillig & Lehmann, 2020*) ecosystems (*Prokić et al., 2021*), mirroring the trends in MP research across different ecosystems. Terrestrial ecosystems, crucial for biodiversity and human well-being (*Baho, Bundschuh & Futter, 2021*), act as significant repositories for MPs (*Büks & Kaupenjohann, 2020*; *Evangeliou et al., 2020*), accumulating 4–23 times more MPs than aquatic environments (*Horton et al., 2017*). Hence, MPs pose an emerging threat to terrestrial ecosystems (*de Souza Machado et al., 2018*), particularly in developing countries (*Zhang et al., 2020a*).

Terrestrial organisms, especially, could face levels of plastic pollution that potentially alter the fundamental aspects of physiological and ecosystem processes (*de Souza Machado et al., 2018*). Despite increasing interest in exploring MP occurrences and their effects on organisms in terrestrial ecosystems, with a higher percentage of studies focusing on invertebrates (especially soil invertebrates) than vertebrates (*Prokić et al., 2021*), research on MP contamination in terrestrial organisms, particularly vertebrates, remains scarce (*Baho, Bundschuh & Futter, 2021*; *Prokić et al., 2021*).

The patterns of MP research in Thailand reflect global trends, with studies primarily concentrating on various ecosystems including marine (*Ruangpanupan et al., 2022*), estuarine (*Chinfak et al., 2021*), wetland (*Sarin & Klomjek, 2022*), rice fields (*Maneechan & Prommi, 2022*), mangroves (*Pradit et al., 2022*), and freshwater ecosystems (*Thamsenanupap, Tanee & Kaewsuk, 2022*; *Tee-hor, Nitiratsuwan & Pradit, 2023*). However, none of the reserch was done in soil and terrestrial ecosystems. Despite this broad coverage, there is a significant gap in research on MP accumulation in terrestrial wildlife, with the majority of studies focusing on invertebrates and fish in freshwater environments. These studies have examined insects (*Thamsenanupap, Tanee & Kaewsuk, 2022*), gastropods (*Jitkaew et al., 2023*; *Yasaka et al., 2022*), and freshwater shrimps (*Tee-hor, Nitiratsuwan & Pradit, 2023*; *Reunura & Prommi, 2022*; *Tongnunui et al., 2022*). Among terrestrial vertebrates, the research has been limited to freshwater fish (*Kasamesiri & Thaimuangphol, 2020*; *Seetapan & Prommi, 2023*), with no studies conducted within protected areas (PAs), despite their critical role in biodiversity conservation (*Naughton-Treves, Holland & Brandon, 2005*).

Addressing the paucity of research on MP contamination in terrestrial vertebrates beyond freshwater fish in Thailand, this study aims to quantify potential MP contamination by analyzing the type, color, and size of MPs in terrestrial vertebrates within and outside PAs in Western Thailand. By examining MPs and their associated biophysical accumulation factors, this research intends to establish a baseline for monitoring plastic pollution in terrestrial vertebrates and to inform the development of effective strategies to mitigate the potential threat of plastic pollution to biodiversity in terrestrial ecosystems, especially within protected areas.

# MATERIALS AND METHODS

## Study area

This study was conducted in Kanchanaburi province, western Thailand. Figure 1 illustrates the locations where carcasses and tadpoles were collected. The human-dominated landscape encompasses forested areas both within and outside PAs and various land-use types outside PAs, primarily agricultural lands and human settlements. The PAs include six national parks (Erawan, Khao Laem, Khuean Srinagarindra, Lam Khlong Ngu, Sai Yok, and Thong Pha Phum) and two wildlife sanctuaries (Salak Phra and Thung Yai West). The population in this area was approximately 646,035, with a density of 114.41 persons/km² (*Department of Provincial Administration (DOPA), 2022*). The Department of National Parks, Wildlife and Plant Conservation of Thailand provided
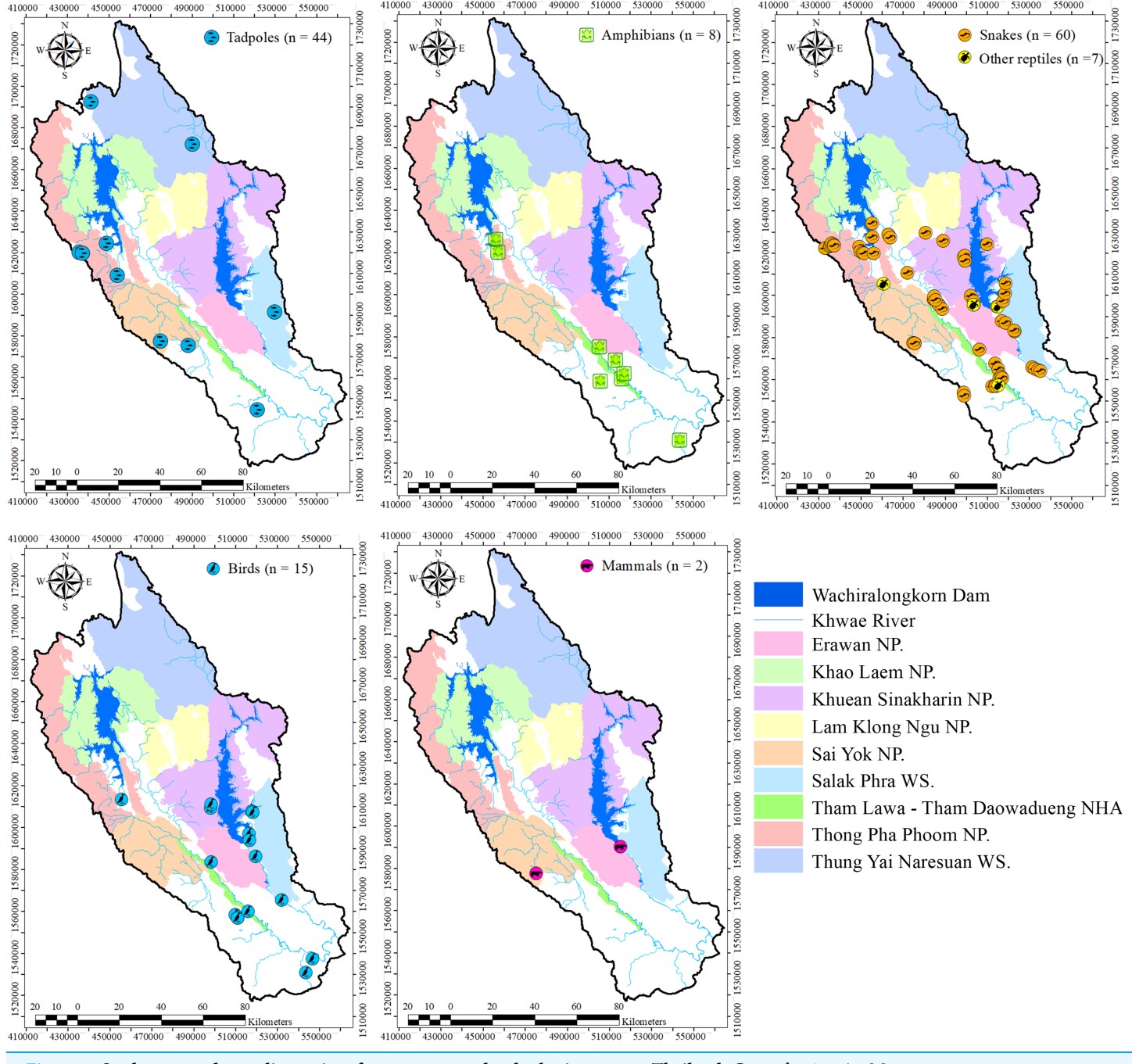

**Figure 1 Study area and sampling points for carcasses and tadpoles in western Thailand.** Created using ArcMap.

a permission for fieldwork in Thai protected areas and for collecting animal carcasses and tadpoles for research (ID#0907.4/17863-26-Aug-2020).

## Sample collection

This research was approved by Animal Care and Use for Scientific Research Kasetsart University ID#ACKU63-*ETC*-001. Road-killed carcasses of amphibians, reptiles, birds,

and mammals in good condition were collected during a survey conducted at a slow speed (below 30 km/h) on a motorcycle. All of them were found on roads. All collected carcasses were fresh, birds had complete plumage (*Deoniziak et al., 2022*), mammals had full fur, and there were no external wounds on any of the animals. Each carcass was wrapped in foil and then sealed in a zip-lock bag. For tadpoles, five individuals of the same species were opportunistically collected at each location and placed in a glass bottle. The locations of the carcasses and tadpoles were recorded using a GPS device (GPS 60csx: Garmin). All samples were stored in an ice cooler at 4 °C during the fieldwork and then transferred to a refrigerator at field stations and subsequently to a freezer at −20 °C in the laboratory for further analysis. It is important to note that whole-body cooling of organisms followed by freezing is considered a humane method of euthanasia for small ectothermic species (less than 4 g: *Lillywhite et al., 2017*).

## Sample preparation in laboratory

Each frozen carcass and tadpole was thawed at room temperature, its weight recorded in grams (to 0.01 g precision), and sizes measured according to standard morphological methods. Each tadpole were washed with deionized water and submerged its whole body in 30% $H_2O_2$ for full digestion for 72 h (*Hu et al., 2018*). The extraction method for MPs in terrestrial carcasses was adapted from *van Franeker & Meijboom (2002)* and *Mathalon & Hill (2014)*. Each carcass was washed with filtered water before the dissection of the gastrointestinal tract. The gastrointestinal tract of each individual was submerged in 10–30 mL of 4 M potassium hydroxide (KOH; 224 g/L) in a glass container under controlled temperature below 60 °C. Then, 5 mL of 35% $H_2O_2$ was added at room temperature to the same container, covered with a petri dish or aluminum foil, and left in the dark for complete tissue digestion for at least 24 h. This method provided a high recovery rate of MPs (*Munno et al., 2017*).

KOH was used for its effectiveness and broad application in digesting tissue and reducing greasy tissue fractions (*Nguyen et al., 2019*), and because polymers (polyethylene, PE, and polypropylene, PP) exhibit resistance to KOH (*Lusher et al., 2017*). Finally, the solutions from the digested tadpoles and carcasses were filtered through a vacuum filtration system using 1.2 µm GF/C filter paper and then stored in lidded glass petri dishes to dry at 50 °C for 4 h.

## Microplastic detection and classification

Potential MPs were visually inspected under a stereomicroscope at 40x magnification using a ZEISS Stemi 508. The MP pieces were sorted, measured for size, classified by type and colors, and photographed. MP classification was adapted from *Wang et al. (2017)*, describing fibers, films, foams, fragments, and pellets based on eight size classes: very small (≤0.05 mm), small (>0.05–0.5 mm), slightly small (>0.5–1 mm), moderate (>1–2 mm), slightly large (>2–3 mm), large (>3–4 mm), very large (>4–5 mm), and debris (>5 mm).

The polymer types of MPs were classified by subsampling 30% of the potential MPs (*Mistri et al., 2021*) for analysis using a micro-Fourier transform infrared

spectrophotometer (μFT-IR; Spotlight 200i-FT-IR microscopy system; PerkinElmer; Waltham, USA). Reflection mode with absorbance spectra wavenumbers in the range 400–4,000 cm[1] was used to compare the spectra of each potential MP to a reference library. Only MPs with over 60% similarity to the standard reference were reported (*Lusher, McHugh & Thompson, 2013*).

### Contamination control

The researchers wore lab coats to reduce the likelihood of contamination. Blank tests were conducted with $H_2O_2$, distilled water, KOH solution and in the laboratory air, parallel to the tissue disintegration. Each blank sample was extracted and detected MPs using the same procedures as above. Less than one MP per individual (0–2) residue indicated a very low level of contamination.

### Landscape characteristics and factors determining microplastic accumulation

Seventeen landscape factors were acquired from the GIS database: location (inside *vs.* outside PAs and headwater-middle-downstream), watershed class, slope, elevation, rainfall, temperature, land-use type, proximity to a village, stream, main road, local road, or tourist site, industrial site, landmark, garbage dump, and the number of households in the nearest village to specimen occurrence. Animal body weight (g) was included as a factor for analysis.

### Statistical analysis

All statistical analyses were conducted using R statistical software (version 4.2.3; *R Core Team, 2023*). The quantities of potential MPs were calculated per individual (MP.ind$^{-1}$) and per weight of sample (MP.g$^{-1}$), and values were presented as mean ± standard deviation (SD) along with the frequency of occurrence. A chi-square test was used to assess the qualitative associations of potential MP occurrences among vertebrate groups. Normality was tested using the Shapiro-Wilk test. For assessing homoscedasticity, the Mann-Whitney U test and the Kruskal-Wallis rank sum test, followed by a Bonferroni *post-hoc* test, were found to be more suitable for comparing MPs across size classes, colors, morphologies, and locations inside *versus* outside PAs, as well as among different PAs. A generalized linear model (GLM) with a negative binomial distribution was developed to examine the factors influencing MP occurrence. The spatial distribution of potential MP occurrences was illustrated using map algebra in ArcMap 10.3 software, employing the results from the GLM.

## RESULTS

A total of 136 samples from 48 species were collected, comprising 92 specimens of 43 vertebrate species across 24 families and eight orders, and 44 specimens of six tadpole species from six families within one order. Among these were samples of the endangered *Trimeresurus kanburiensis* and the vulnerable *Ophiophagus hannah*, with the remaining species classified as of least concern by the *IUCN (2023)*. The majority of the specimens

**Table 1 Numbers of species, specimen, occurrence, numbers of potential microplastics and average numbers per specimen and per body weight (g.) in each animal group.**

| Vertebrate | No. of species | No. specimens (% of all) | MP occurrence (%) | Total MP | MP.ind$^{-1}$ | MP.g$^{-1}$ |
|---|---|---|---|---|---|---|
| Amphibians | 4 | 8 (5.88%) | 5 (62.5%) | 34 | 4.25 ± 7.50 | 0.04 ± 0.09 |
| Snakes | 20 | 60 (44.12%) | 55 (91.67%) | 211 | 3.52 ± 3.20 | 0.05 ± 0.09 |
| Reptiles | 6 | 7 (5.15%) | 5 (71.43%) | 9 | 1.29 ± 1.25 | 0.02 ± 0.03 |
| Birds | 11 | 15 (11.03%) | 13 (86.67%) | 41 | 2.73 ± 3.22 | 0.03 ± 0.05 |
| Mammals | 2 | 2 (1.47%) | 2 (100%) | 4 | 2.00 ± 0.00 | 0.01 ± 0.004 |
| Tadpoles | 6 | 44 (32.35%) | 31 (70.46%) | 88 | 2.00 ± 3.13 | 12.88 ± 30.79 |
| All groups | 48 | 136 | 111 | 387 | 2.86 ± 3.52 | 4.20 ± 18.39 |

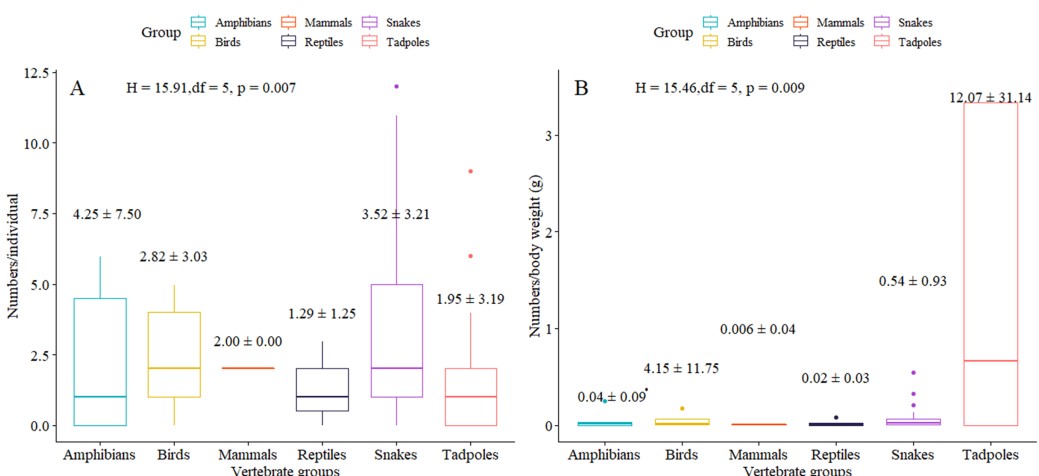

**Figure 2 Average number of microplastics (± standard deviation) found in each animal group by individual specimen (A) and by body weight of specimen (B).**

were snakes (44.12%), followed by tadpoles (32.35%), birds (32.35%), amphibians (5.88%), other reptiles (5.15%), and mammals (1.47%).

Over 60% of the specimens across each animal group were contaminated by 387 potential MPs, averaging 2.86 ± 3.52 MP.ind$^{-1}$ and 4.20 ± 18.39 MP.g$^{-1}$. Amphibians exhibited the highest MP.ind$^{-1}$ rates, while tadpoles had the highest MP.g$^{-1}$ levels (Table 1), with statistically significant differences observed among groups, especially between snakes and tadpoles (Fig. 2).

MP occurrences were slightly over 60% in all animal groups, ranging from 2 to 4.25 MP.ind$^{-1}$. Snakes showed the highest susceptibility to MP contamination (91.67%), followed by birds (86.67%), other reptiles (71.43%), tadpoles (70.46%), and amphibians (62.5%). However, in terms of both per individual and weight, snakes were the second-most susceptible to higher MP accumulations (3.52 ± 3.20 MP.ind$^{-1}$ or 0.05 ± 0.09 MP.g$^{-1}$) after amphibians (4.25 ± 7.50 MP.ind$^{-1}$) and tadpoles (12.88 ± 30.79 MP.g$^{-1}$), respectively. Table S1 details the quantities of MPs by each species of terrestrial animals.

## Microplastic occurrence by type, color, and size

Fibers were the most abundant MPs (77.00%, $2.19 \pm 2.59$ MP.ind$^{-1}$, $0.07 \pm 0.08$ MP.g$^{-1}$), followed by fragments (22.74%, $0.63 \pm 1.69$ MP.ind$^{-1}$, $0.02 \pm 0.05$ MP.g$^{-1}$), foam (0.52%, $0.02 \pm 0.12$ MP.ind$^{-1}$, $0.001 \pm 0.004$ MP.g$^{-1}$), and film (0.26%, $0.01 \pm 0.09$ MP.ind$^{-1}$, $0.0002 \pm 0.003$ MP.g$^{-1}$). There was a significant association between animal groups and MP types ($\chi2 = 63.52$, df = 20, $p < 0.01$), especially between amphibians and fibers ($p = 0.03$).

Most MPs were blue (41.86%; $1.19 \pm 2.03$ MP.ind$^{-1}$, $0.003 \pm 0.004$ MP.g$^{-1}$), followed by black (19.64%; $0.56 \pm 1.11$ MP.ind$^{-1}$, $0.001 \pm 0.002$ MP.g$^{-1}$) and white (7.75%; $0.22 \pm 0.82$ MP.ind$^{-1}$, $0.0005 \pm 0.002$ MP.g$^{-1}$), with the remaining nine colors each comprising less than 5%. Each animal group was significantly associated with MP colors ($\chi2 = 99.37$, df = 55, $p < 0.01$), especially birds with brown ($p < 0.01$). Only the blue color showed significant differences among animal groups, both per individual and per weight (H = 12.83, df = 5, $p = 0.03$), notably between snakes and tadpoles ($p = 0.01$).

MPs sized from 43.46 to 2,504.01 μm showed large proportions of small (30.23%, $0.86 \pm 1.94$ MP.ind$^{-1}$, $0.003 \pm 0.006$ MP.g$^{-1}$), moderate (27.91%, $0.79 \pm 1.16$ MP.ind$^{-1}$, $0.003 \pm 0.004$ MP.g$^{-1}$), and slightly small (22.22%, $0.63 \pm 1.04$ MP.ind$^{-1}$, $0.002 \pm 0.003$ MP.g$^{-1}$) sizes. Other sizes accounted for less than 10%. Significant differences were observed between animal groups and MP sizes ($\chi2 = 68.66$, df = 40, $p < 0.01$), especially mammals and large sizes ($p = 0.02$). Only small MPs (>0.5–1 mm) significantly differed by individual and by weight (W = 2,598.5, $p < 0.01$). Significant size differences with MP occurrences existed for very small (<0.05 mm: H = 16, df = 5, $p < 0.01$), small (H = 14.56, df = 5, $p = 0.01$), and large (3,000–4,000 μm: H = 12.33, df = 5, $p = 0.03$) between snakes and amphibians. Figure 3 displays the quantities of potential MPs classified by vertebrate groups into types, colors, and sizes, with additional details in the Supplemental Data. Table S2 showed quantities of MPs by types, colors, and sizes in terrestrial animals.

## Microplastic occurrence inside and outside protected areas

More than 50% of all specimens collected showed evidence of potential MPs, with a higher incidence in carcasses found outside PAs (61.50%) compared to those inside PAs (38.50%). However, the differences in MP accumulation per specimen (W = 1,866, $p = 0.05$) and per weight (W = 1,896.5, $p = 0.07$) were not statistically significant, as indicated in Table 2.

In areas outside PAs, the prevalence of fibers and fragments was 34.44% and 43.64% higher, respectively, than inside PAs. Films and foams were absent inside PAs. The occurrence of fibers was significantly different between inside and outside PAs (W = 1,791, $p = 0.02$), in contrast to fragments (W = 2,206, $p = 0.53$), films (W = 2,277, $p = 0.32$), and foams (W = 2,242.5, $p = 0.15$).

All 12 colors of MPs were identified both inside and outside PAs, with greater quantities observed outside, except for green and red. Specifically, blue and black MPs were found to be 70% and 10% more prevalent, respectively, outside PAs than inside. The variation in MP occurrence by color between inside and outside PAs was significantly different only for purple (W = 2,005, $p = 0.02$).

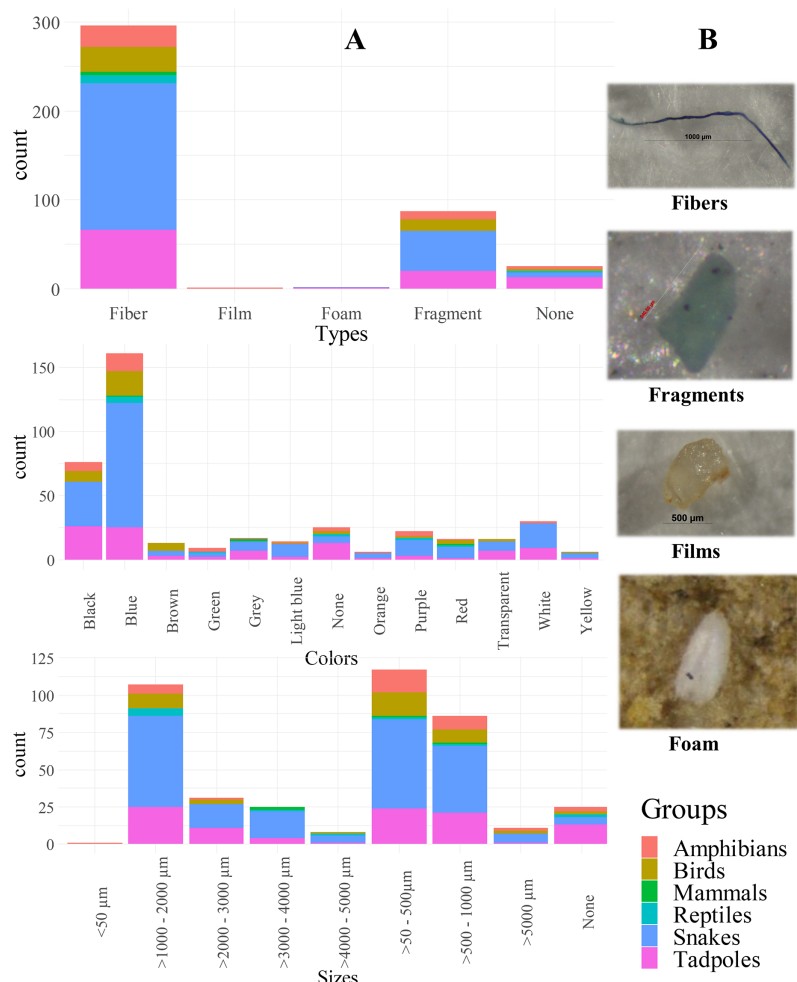

**Figure 3** Proportion of specimen with the numbers of microplastic contamination characterized by animal groups in association with types, colors, and sizes (A). None referred to specimen without MP contamination and examples of MP by types (B).

MPs of all sizes were more commonly found outside than inside PAs, with the exception of the very large size category, which was equally represented in both areas. Small MPs were the most frequently observed category both inside and outside PAs; however, their occurrence was 41.89% higher outside PAs. Significant differences in MP occurrence between inside and outside PAs were noted only for the slightly small-sized MPs (>0.5–1 mm) in terms of both MP per individual (MP.ind$^{-1}$) and MP per gram (MP.g$^{-1}$) (W = 1,865.5, $p = 0.02$).

## Factors affecting microplastic occurrence

To examine the factors influencing microplastic occurrence in terrestrial wildlife and tadpoles, three models were evaluated, each with a different combination of specimens: (1) all terrestrial vertebrates and tadpoles, (2) only snakes, and (3) only tadpoles. Microplastic occurrences in all animal groups and specifically in snakes were found to be influenced by several factors, including body weight, proximity to a local road, proximity to a garbage

Table 2 Numbers of specimens of vertebrate species studied on frequency occurrences and abundance of microplastics in each group of vertebrates and tadpoles, compared between inside and outside protected areas.

| Group | PA site | No. Sp. | No. specimen | MP occurrence (%) | Total MP | MP.ind$^{-1}$ | MP.g$^{-1}$ |
|---|---|---|---|---|---|---|---|
| Amphibians | Inside | 2 | 2 | 1 (50.00%) | 6 | 3.00 ± 4.24 | 0.013 ± 0.018 |
| | Outside | 4 | 6 | 4 (66.67%) | 28 | 4.67 ± 8.62 | 0.05 ± 0.10 |
| Snakes | Inside | 16 | 29 | 26 (89.66%) | 98 | 3.38 ± 2.96 | 0.03 ± 0.04 |
| | Outside | 14 | 31 | 29 (93.55%) | 113 | 3.65 ± 3.46 | 0.07 ± 0.12 |
| Lizards | Inside | 5 | 6 | 4 (66.67%) | 8 | 1.33 ± 1.37 | 0.02 ± 0.03 |
| | Outside | 1 | 1 | 1 (100.00%) | 1 | 1 | 0.001 |
| Birds | Inside | 5 | 5 | 4 (80.00%) | 11 | 2.20 ± 1.92 | 0.07 ± 0.06 |
| | Outside | 6 | 10 | 9 (90.00%) | 30 | 3.00 ± 3.77 | 0.02 ± 0.02 |
| Mammals | Inside | 1 | 1 | 1 (100.00%) | 2 | 2.00 | 0.009 |
| | Outside | 1 | 1 | 1 (100.00%) | 2 | 2.00 | 0.003 |
| Tadpoles | Inside | 4 | 26 | 14 (53.85%) | 24 | 0.92 ± 1.13 | 1.98 ± 6.06 |
| | Outside | 4 | 18 | 17 (94.44%) | 64 | 3.56 ± 4.31 | 28.63 ± 43.59 |
| All groups | Inside | 33 | 69 | 50 (72.46%) | 149 | 2.16 ± 2.45 | 0.77 ± 3.80 |
| | Outside | 28 | 67 | 61 (91.05%) | 238 | 3.55 ± 4.25 | 7.73 ± 25.54 |

Table 3 Factors affecting MP quantity in all combined terrestrial vertebrates (amphibians, snakes, other reptiles, birds, mammals, and tadpoles) analyzed using generalized linear model (GLM: negative binomial).

| Parameter | Estimate | Standard error | Z | P |
|---|---|---|---|---|
| Intercept | 1.79 | 0.23 | 7.73 | $1.09 \times 10^{-14}$ |
| Body weight (g) of animals | $2.93 \times 10^{-4}$ | $8.15 \times 10^{-5}$ | 3.60 | $3.17 \times 10^{-4}$ |
| Proximity to local road | $-5.50 \times 10^{-5}$ | $1.61 \times 10^{-5}$ | −3.41 | $6.59 \times 10^{-4}$ |
| Proximity to garbage dump | $-4.89 \times 10^{-5}$ | $1.77 \times 10^{-5}$ | −2.76 | $5.83 \times 10^{-3}$ |
| Landuse (Agriculture as reference) | | | | |
| Forest | −0.59 | 0.24 | −2.45 | 0.01 |
| Human settlement | −0.36 | 0.23 | −1.55 | 0.12 |

Note: Remark: Null deviance = 186.99 on 135 df, Residual deviance = 142.13 on 130 df, AIC = 574.67, Theta = 1.84, SE = 0.39, 2x log-likelihood = −560.674.

Table 4 Factors affecting MP quantity in snakes analyzed using generalized linear model (GLM: negative binomial).

| Parameter | Estimate | Standard error | Z | P |
|---|---|---|---|---|
| Intercept | 2.00 | 0.23 | 8.76 | $<2 \times 10^{-16}$ |
| Snake body weight (g) | $3.0 \times 10^{-4}$ | $7.59 \times 10^{-5}$ | 3.95 | $7.9 \times 10^{-5}$ |
| Proximity to garbage dump | $-5.33 \times 10^{-5}$ | 0.24 | −3.02 | 0.003 |
| Landuse (Agriculture as reference) | | | | |
| Forest areas | −0.74 | 0.24 | −3.11 | 0.002 |
| Human settlement | −0.64 | 0.24 | −2.63 | 0.009 |
| Proximity to local road | −6.82 | $2.11 \times 10^{-5}$ | −3.23 | 0.001 |

Note: Remark: Null deviance = 162.60 on 101 df, Residual deviance = 107.78 on 96 df, AIC = 421.31, Theta = 2.73, SE = 0.81, 2x log-likelihood = −407.32.

**Table 5 Factors affecting MP quantity in tadpoles analyzed using generalized linear model (GLM: negative binomial).**

| Parameter | Estimate | SE | Z | P |
|---|---|---|---|---|
| Intercept | −0.28 | 0.24 | −1.19 | 0.24 |
| Number of households at sampling location | 0.002 | $3.20 \times 10^{-4}$ | 5.43 | $5.6 \times 10^{-8}$ |

**Note:** Remark: Null deviance = 78.46 on 41 df, Residual deviance = 42.53 on 40 df, AIC = 140.69, Theta = 2.85, SE = 0.81, 2x log-likelihood = −407.32.

dump, and land-use type, as shown in Tables 3 and 4. In contrast, for tadpoles, only the number of households in the vicinity was significantly associated with microplastic occurrence, as detailed in Table 5. Figure 4 provides a spatial representation of the microplastic risk distribution.

## Polymer types of microplastics

The investigation identified four thermoplastic polymers: polyethylene (PE), polyethylene terephthalate (PET), polypropylene (PP), and polyvinylidene chloride (PVDC), alongside one thermosetting plastic, polyester (PES). Additionally, various natural and synthetic materials were found, including rayon, azlon, and cotton. Notably, combinations of synthetic and natural fibers were detected, such as cotton mixed with polyurethane (PU), and rayon combined with either PES, PU, or silicone rubber.

## DISCUSSION

Our study has highlighted MP accumulation in terrestrial vertebrates and tadpoles in Thailand, both within and outside protected areas. This contributes significant insights into MP contamination, an area previously less explored in terrestrial and freshwater contexts compared to marine environments, as evidenced by studies on marine vertebrates (*Park et al., 2023*) and freshwater fish (*Kasamesiri & Thaimuangphol, 2020*; *Seetapan & Prommi, 2023*).

## Occurrence and quantity of microplastics in terrestrial wildlife

We observed MP occurrence in over 60% of all specimens, with concentrations ranging from 2.00 to 4.25 MPs per individual. Compared to other studies on terrestrial wildlife and tadpoles, our findings generally report lower levels of MP, except in the case of reptiles. The comparatively modest quantities of MPs detected in terrestrial wildlife from western Thailand may not fully represent the actual levels of MP contamination, potentially overlooking MPs in other organs like the lungs and livers of birds (*Tokunaga et al., 2023*; *Lim, 2021*) or bird tissues (*Tanaka et al., 2013*). Methodological differences and varying species studied limit direct comparisons, but our research addresses some knowledge gaps regarding MP presence in terrestrial vertebrates from Thailand and Southeast Asia.

## Types, colors and sizes of microplastics in terrestrial wildlife

Our analysis identified small (0.05–0.5 mm) blue fibers as the predominant MP type and size in terrestrial wildlife, mirroring findings in snakes from museum collections (*Gül et al., 2022*) and frogs from the Bengal delta (*Shetu, Parvin & Tareq, 2023*). Similar observations

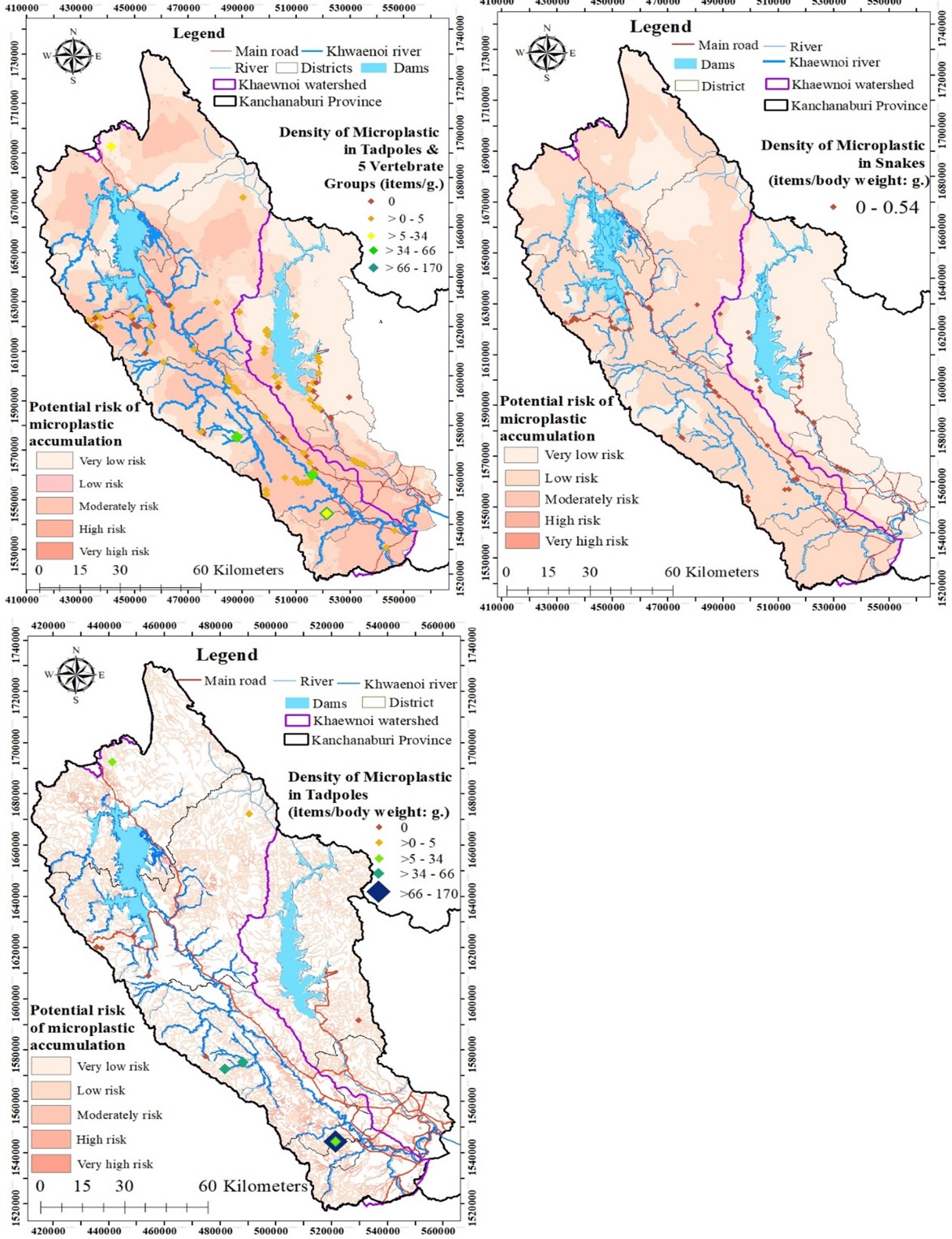

**Figure 4** Maps of microplastic occurrence for all groups of terrestrial wildlife and tadpoles (A), only snakes (B), and only tadpoles (C). Created using ArcMap.

were made in birds of prey and frogs (*Carlin et al., 2020*; *Tatlı, Altunısık & Gedik, 2022*; *Pastorino et al., 2023*), with fibers less than 0.5 mm also noted in birds (*Deoniziak et al., 2022*) and tadpole species (*Hu et al., 2018*). Fibers, particularly prevalent in natural settings, pose inhalation risks to wildlife and can be ingested but are often expelled through the gut (*Koutnik et al., 2021*; *Gasperi et al., 2018*; *da Costa Araújo et al., 2020*). The abundance of blue MPs suggests a visual attraction, possibly due to animals mistaking these for food, with blue's visibility in freshwater, soil, and air also playing a role (*Foekema et al., 2013*; *Thamsenanupap, Tanee & Kaewsuk, 2022*; *Tee-hor, Nitiratsuwan & Pradit, 2023*; *Zhang et al., 2020b*; *Koutnik et al., 2021*). Predominantly small-sized MPs (0.05–0.5 mm) were consistent across different species, likely influenced by factors such as feeding mechanisms, foraging behavior, and environmental degradation processes (*Hu et al., 2018*; *Liu et al., 2023a*; *Zhang et al., 2020b*).

## Polymer types, sources and their impacts on wildlife

This study classified polymer types based on the types and colors of MPs identified. Our findings revealed that PE, PET, PP, PVDC, and PES were the predominant polymers found in terrestrial vertebrates and tadpoles. Mixtures of rayon with either PES, PU, or silicone rubber, and cotton with PU, were also detected. A limitation of our research is that the identified MPs were pooled from all examined carcasses and tadpoles, preventing us from determining the proportion of each polymer type within different vertebrate groups.

The mechanisms distributing these polymers to terrestrial vertebrates or tadpoles could vary. Sources of MPs include littering, open dumps, improperly managed landfills, atmospheric deposition, runoff from roads, agriculture, and settlements (*de Souza Machado et al., 2018*; *Horton et al., 2017*), as well as from fishing gears (*Ruangpanupan et al., 2022*) and plastic bags used in checkdam construction. Animals may ingest PE, PP, and PET from plastic bags, bottles, ropes, or fishing gears, commonly used in packaging (*Wang, Zhao & Xing, 2021*) and are prevalent in daily items (*Liu et al., 2020*), especially in Thailand (*ONEP, 2023*), and are frequently encountered in urban-rural gradients (*Kunz et al., 2023*), including our study area. PE and PP might be released from personal care products containing microbeads (*Napper et al., 2015*) and fishing gears (*Chinfak et al., 2021*). Food packaging, plastic containers, and pipes are sources of PP and PET (*Chinfak et al., 2021*). Road runoff could introduce PE (*Piñon-Colin et al., 2020*), PP (*Lange et al., 2022*), and PES (*Rosso et al., 2022*), as carcasses were collected from roadkill, and most collection sites were road-accessible. Another significant source of PES, other partially synthetic fibers (azlon and rayon), and natural fiber (cotton) in terrestrial wildlife may be clothing (*Thrift et al., 2022*) and laundry activities (*De Falco et al., 2018*). PVDC may originate from plastic wraps, household products, and filters (*Plastic Europe, 2020*). PU foam, often used in Thailand for home furnishings (*Ounjai et al., 2020*), and buoys in fish cage culture (*Ruangpanupan et al., 2022*), also contributes to the presence of PU.

The polymer types detected in terrestrial vertebrates and tadpoles in our investigation closely mirrored those found in other research on various biotic groups in Thailand. Notably, PE, PP, and PET emerged as the prevalent polymer types of microplastics

accumulated in wild fauna across Thailand. For instance, PE, PET, and PP were identified in pelagic, demersal, and benthopelagic freshwater fish (*Seetapan & Prommi, 2023*). PE and PP were reported in mussels, clams (*Chinfak et al., 2021*), two fish species (*Barbonymus altus*, *Laides longibarbis*), and two snail species (*Filopaludina martensi*, *Pomacea canaliculata*) (*Yasaka et al., 2022*). PET was detected in rice field-dwelling *Pantala sp.* (*Maneechan & Prommi, 2022*), and two snail species (*F. sumatrensis speciosa*, *P. canaliculata*) from the U-Taphao River (*Jitkaew et al., 2023*). PP contamination was observed in *Pantala sp.* from rice fields (*Maneechan & Prommi, 2022*) and the two snail species (*Jitkaew et al., 2023*). Rayon was found in mussels, clams (*Chinfak et al., 2021*), *P. canaliculata, F. sumatrensis* (*Jitkaew et al., 2023*), and Giant freshwater prawns (*Macrobrachium rosenbergii*) (*Tee-hor, Nitiratsuwan & Pradit, 2023*). PU contamination was noted in the two fish species and the two snail species (*Yasaka et al., 2022*), while Giant freshwater prawns were also contaminated with cotton (*Tee-hor, Nitiratsuwan & Pradit, 2023*), PE (*Reunura & Prommi, 2022*), and PES (*Tongnunui et al., 2022*).

Research conducted in Thailand has not yet explored the morphological and physiological effects of different polymer types on specific wildlife species. Amphibians and tadpoles are frequently used as models for toxicity testing (*da Costa Araújo et al., 2020*; *Prokić et al., 2021*). Among various polymers, PE is the most commonly examined in toxicological studies. However, current knowledge is insufficient to definitively conclude that PE has more adverse effects on any group of terrestrial vertebrates or tadpoles than on others. Studies by *da Costa Araújo et al. (2020)* have shown that PE exposure in tadpoles of *Physalaemus cuvieri* results in minor external morphological changes, an increase in melanophores and pigmentation rate, and accumulation in organs such as the gills, gastrointestinal tract, liver, muscle tissues of the tail, and blood. In mice, PE accumulation has been observed in the intestine, liver, and kidney, causing intestinal inflammation at high concentrations (*Liu et al., 2023b*). Furthermore, exposure to PE has been linked to reduced body weight, fecundity, metabolism, and alterations in the weight and sex ratio of offspring in mice (*Park et al., 2020*). The impact of PE, however, may vary with its grade, affecting its fate, behavior, and ecological impact differently (*Andrady, 2017*).

Exposure of *Xenopus laevis* tadpoles to polyester MP fibers, at concentrations of 10 or 50 µg/mL, led to significant abnormal gut coiling (*Bacchetta et al., 2021*). Similarly, *Rana sylvatica* tadpoles exposed to PES showed an increased susceptibility to infection by trematodes in natural settings (*Buss, Sander & Hua, 2022*). Terrestrial vertebrates exposed to PVDC might experience apoptosis and morphological damage to cell membranes, as seen in mouse primary liver cells, though this effect is not observed with PE (*Yamamoto et al., 2020*). Meanwhile, PP has shown no significant toxicological effects on mortality, body weight, organ histology, hormone levels, fertility, hatch rates, or eggshell strength in studies. However, endocrine effects have been noted in Japanese Quail (*Coturnix japonica*) fed bio-fouled PP pellets in a laboratory setting (*Roman et al., 2019*).

## Factors affecting potential microplastic accumulation in terrestrial vertebrates

Several factors influence MP accumulation in terrestrial vertebrates, which are pertinent to human activities. We found that the type, size, and color of MPs were similar among water, soil, sediment, and terrestrial carcasses, although not significantly associated. This result is in line with *Hu et al. (2018)*, who reported that the abundance, shape, and polymer distribution of MPs in tadpoles resembled those found in water.

Animal body weight also positively correlated with the MP load in the combined samples of terrestrial vertebrates and snakes. However, this correlation did not extend to tadpoles. This finding contradicts *Gül et al. (2022)*, who found a negative correlation between MP load and tadpole weight, and *Hu et al. (2018)*, who reported a positive correlation between tadpole length and MP load. Such discrepancies between studies suggest the need for better research design to understand the effects of animal size on MP accumulation.

Our results highlighted that proximity to human-related landscapes, such as roads, land use types, and the number of households, had a statistically significant influence on MP accumulation. This was evidenced by higher MP quantities in terrestrial wildlife outside PAs compared to inside PAs. We also found that poorly managed garbage dumps were another source of MPs, as they could attract vertebrates to feed on food waste and accidentally ingest contaminated plastic debris (*Teampanpong, 2021*), along with MPs degraded from plastic debris. The influence of the number of households on MP quantities in tadpoles and the importance of proximity to garbage dumps and land use types confirmed the linkage between MP quantities and human activities.

Furthermore, our results showed for the first time that lower levels of MP accumulation were found in terrestrial vertebrates living in PAs than in those living outside PAs. This finding, while anticipated, was confirmed by the human-related influence on the quantities of MP accumulation, as *Kutralam-Muniasamy et al. (2021)* reported MP ingestion by over 50% of organisms living in PAs. Less human use and stringent measures to control plastic waste in Thai PAs might further lower MP accumulation compared to areas outside PAs. Locations outside PAs and multiple-use areas within PAs accumulate more MPs than restricted zones (*Nunes et al., 2023*).

It is assumed that the source of MP accumulation in terrestrial vertebrates originated from two environmental channels. The first was through the animals feeding, both intentionally and accidentally, not only on food or contaminated prey (*Thrift et al., 2022*) but also ingesting contaminated MPs in water, soil, or sediment (*English et al., 2015*; *Holland, Mallory & Shutler, 2016*) and from using plastic for nesting and encountering it during burrowing (*Ayala et al., 2023*). However, our results showed no relationship between MPs in terrestrial vertebrates and in water, soil, and sediment. The second source was through trophic transfer in the food web (*Cole et al., 2011*). For example, rainfall runoff from outside PAs may carry MPs into PAs (*Brahney et al., 2020*; *Forster, Wilson & Tighe, 2023*), and MPs in PAs can come from the clothing, footwear, and food packaging of tourists (*Forster, Wilson & Tighe, 2023*).

Although MP accumulation in terrestrial wildlife in Thailand appeared to be at levels generally low regarding causing serious harm, preventive measures should be implemented on the use and transfer of plastics within PAs, especially in restricted zones, as MPs represent an emerging threat to biodiversity from MP degradation to toxic chemical derivatives (*Zang et al., 2020*). Actions are necessary to manage human activities as sources of MPs, particularly through standardizing solid waste management at garbage dumps. Stricter regulations on plastic use and disposal are imperative, especially in remote areas. It is crucial for the Thai government to take proactive measures to improve plastic use and waste management across the country. Additionally, establishing a monitoring plan for MP pollution in wildlife and terrestrial ecosystems, particularly in PAs, is essential. Further research is needed on the ecotoxicological impacts of MPs on terrestrial wildlife and tadpoles to identify indicator species of microplastic pollution. More stringent policies on prohibiting single-use plastics and promoting plastic reuse, the use of alternative materials, and recycling (*Lim, 2021*) should be enforced not only in Thailand but globally. These insights will be crucial in developing effective strategies for mitigating the effects of MPs on biodiversity and ecosystem health.

## CONCLUSIONS

Our study delved into the prevalence of microplastics (MPs) in terrestrial wildlife and tadpoles, uncovering contamination across various animal groups of terrestrial wildlife in Thailand and is among the few studies conducted in Southeast Asia. Despite the potentially low abundance of MPs, mostly small (0.05–0.5 mm) blue fibers, a high occurrence rate of 60% among all specimens even within protected areas located in remote regions with minimal human activity, raises conservation concerns. Notably, the contamination of MPs in globally endangered and vulnerable species and amphibians, which exhibited the highest levels of contamination underscores the potential threat MPs might pose to already imperiled populations, even within protected areas. Analyses of MPs across different wildlife groups revealed distinct associations between human activities, highlighting the complex role human activities play in shaping sources and distribution of MPs in terrestrial wildlife, both inside and outside protected areas. Our study recommends the use of snakes from road-kills over live capture for non-invasive MP analysis in monitoring efforts. Further research is necessary to better understand the long-term consequences of MP contamination and the ecotoxicological impacts of MPs on terrestrial vertebretes and tadpoles.

## ACKNOWLEDGEMENTS

We thank the Department of National Parks, Wildlife and Plant Conservation of Thailand for permitting the research in Thai protected areas. We appreciate the field assistance provided for data collection by the park rangers in all the PAs and by the undergraduate and graduate students in the Forest Biology Department at the Faculty of Forestry, Kasetsart University. We appreciate Assistant Professor Dr. Yodchaiy Chuaynkern for identification of amphibian species both mature specimen and tadpoles. We thank Dr. Sampan Thongnuniu for providing support during the fieldwork. We also thank the

laboratory technicians at the Department of Conservation and the Department of Forest Biology at Faculty of Forestry, Kasetsart University for assisting us, even during the Covid-19 lockdown.

### Funding
This work was supported under Research and Innovation Grant 2020 by the National Research Council of Thailand. The funders had no role in study design, data collection and analysis, decision to publish, or preparation of the manuscript.

### Grant Disclosures
The following grant information was disclosed by the authors:
National Research Council of Thailand.

### Competing Interests
The authors declare that they have no competing interests.

### Author Contributions
- Jiraporn Teampanpong conceived and designed the experiments, performed the experiments, analyzed the data, prepared figures and/or tables, authored or reviewed drafts of the article, and approved the final draft.
- Prateep Duengkae conceived and designed the experiments, performed the experiments, analyzed the data, authored or reviewed drafts of the article, laboratory equipment for analysis, and approved the final draft.

### Animal Ethics
The following information was supplied relating to ethical approvals (*i.e.*, approving body and any reference numbers):
Animal Care and Use for Scientific Research Kasetsart University (ACKU63-*ETC*-001).

### Field Study Permissions
The following information was supplied relating to field study approvals (*i.e.*, approving body and any reference numbers):
Department of National Park, Wildlife, and Plant Conservation of Thailand

### Data Availability
The raw data is available in the Supplemental File.

### Supplemental Information
Supplemental information for this article can be found online at http://dx.doi.org/10.7717/peerj.17384#supplemental-information.

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
