# Peer review of "Terrestrial wildlife as indicators of microplastic pollution in western Thailand"

_PeerJ, doi:10.7717/peerj.17384_

## Round 0.1 · original submission · Major Revisions

I am expecting a revised and improved version, following the reviewer's comments.

**Language Note:** The review process has identified that the English language must be improved. PeerJ can provide language editing services - please contact us at [email protected] for pricing (be sure to provide your manuscript number and title). Alternatively, you should make your own arrangements to improve the language quality and provide details in your response letter. – PeerJ Staff

Reviewer 1 ·

Basic reporting

The authors have submitted an interesting and important study to evaluate the penetration of microplastics into terrestrial life. Overall, it seems considerable for publication in PeerJ; however, before it is considered for publication, it must be revised and improved.
I have the following suggestions/comments on this manuscript and urge authors to prepare a revised version.
1. The language of the manuscript needs substantial improvement.
1. “We” should not be used in the manuscript. For instance, the sentence “We collected the corpses of road-killed vertebrates in good conditions and also live tadpoles to investigate their exposure to plastic pollution. We detected MPs with sizes over 50 μm and assessed their occurrence among the animal groups and tadpoles. We collected 136 individuals of 48 vertebrate species,” should be re-written as “The road-killed corpses of the vertebrates and live tadpoles were collected to investigate their exposure to plastic pollution. In total, 136 individuals of 48 vertebrate species were studied, which showed to contain MPs with sizes over 50 μm.”
2. Language should be concise and more meaningful, and explain the manuscript’s story in the sequence of events that actually happened in the lab.

Experimental design

3. Which parts of the bodies contained MPs? Please add the abstract
4. The introduction section should be reorganized and contain 3 paragraphs; the 1st one could be a basic introduction to Plastic and its distribution to life. 2nd one should cover a review of literature that has previously studied aquatic and terrestrial life, what has been explored, and what research gaps are there. How can those be addressed? 3rd paragraph should contain the objectives of the current study, novelty, and significance.
5. Please make sure that methods provide enough details for the readers to reproduce and follow
6. It would be nice if the distribution mechanism is explained with references, and what morphological and physiological effects were caused by specific MPs on life? Which kind of samples were being affected the most badly?

Validity of the findings

7. Conclusion should not simply repeat the results; it should tell what you conclude based on the findings. Please add some recommendations for the policymakers/researchers to control/regulate/mitigate the MP's problem in protected areas.

Reviewer 2 ·

Basic reporting

After a thorough review, I find this work to be interesting and worthy of publication in PeerJ. The manuscript is well-drafted and includes all the necessary components. I suggest some modifications to enhance the quality of the article. I recommend that this work be published after minor revisions.

Experimental design

The research design is appropriate, and the methods are adequately described.

Validity of the findings

- The introduction needs to be divided into four distinct paragraphs. The first paragraph should address the problem statement, the second should discuss the current solution and its limitations, the third should present the proposed solution, and the fourth should outline the aims and objectives of this research paper.

- The authors are encouraged to suggest, based on the findings in this article, an area of research or space that still requires further exploration.

- The authors should provide a brief commentary on the disadvantages of the studied systems. An additional paragraph should be added to offer readers a more in-depth understanding of the subject.

- The conclusion is too generalized and requires modification. It should focus on highlighting only the most valuable findings of the research.

Additional comments

- The entire manuscript should be formatted in accordance with the journal guidelines. There are several spacing and other errors that need correction.

·

Basic reporting

The study investigates microplastic pollution in terrestrial wildlife in western Thailand, exploring its impact on various species within and outside protected areas. Through analyzing road-killed vertebrates and live tadpoles, the research found widespread microplastic presence, differing by animal type, location relative to protected areas, and other factors. This study contributes to our knowledge of microplastic contamination and is useful for better waste management. The article basic reporting is fine. English language should be further refined.

Experimental design

Some aspects need to be clarified in this article as follows:
1- the criteria for selecting animal specimens should be included: why such animals were chosen, and why were tadpoles collected live?
2- How were the carcasses transferred to the laboratory?
3- Please provide the condition of the carcasses where they were found. For instance, were they found in water?
4- Please provide the reason why you collected water, soil, and sediment samples. What is the relationship with the collected animal specimens?
5- Specify the conditions for adding hydrogen peroxide, as it might degrade some microplastics.
6- Add reasons for why sodium chloride was used in the floatation test, as some heavy polymers might not be recovered.
7- Blank replicates should be typical. Please clarify accordingly.

Validity of the findings

The conclusion is not well written; for instance, why should we not neglect microplastic pollution in small amounts? Draw your conclusion from the results.

---

## Round 0.2 · accepted · Accept

Please address the following comments during proofreading!

1- please clarify why snakes obtained could serve as a non-invasive method for monitoring plastic pollution.
2- please add a scale bar for the foam pic in Figure 3.

Reviewer 1 ·

Basic reporting

The authors have incorporated all the suggested changes.

Experimental design

The authors have incorporated all the suggested changes.

Validity of the findings

The authors have incorporated all the suggested changes.

Reviewer 2 ·

Basic reporting

In the revised submission, the authors have effectively addressed the comments and concerns raised by the reviewers regarding their initial submission. Consequently, I recommend publication.

Experimental design

In the revised submission, the authors have effectively addressed the comments and concerns raised by the reviewers regarding their initial submission. Consequently, I recommend publication.

Validity of the findings

In the revised submission, the authors have effectively addressed the comments and concerns raised by the reviewers regarding their initial submission. Consequently, I recommend publication.

·

Basic reporting

1- please clarify why snakes obtained could serve as a non-invasive method for monitoring plastic pollution.
2- please add a scale bar for the foam pic in Figure 3.

Experimental design

No other comments.

Validity of the findings

No other comments.